# PROCEEDINGS A

materials science, computational physics

pulsed DC magnetron sputtering, inert gas clusters, diffusion, blistering

**Author for correspondence:**
Roger Smith
e-mail: r.smith@lboro.ac.uk

# Inert gas bubble formation in magnetron sputtered thin-film CdTe solar cells

Peter Hatton, Ali Abbas, Piotr Kaminski, Sibel Yilmaz[†], Michael Watts, Michael Walls, Pooja Goddard and Roger Smith

Loughborough University, Loughborough LE11 3TU, UK

RS, 0000-0001-8147-431X

Cadmium telluride (CdTe) solar cells are deposited in current production using evaporation-based techniques. Fabricating CdTe solar cells using magnetron sputtering would have the advantage of being more cost-efficient. Here, we show that such deposition results in the incorporation of the magnetron working gas Ar, within the films. Post deposition processing with $CdCl_2$ improves cell efficiency and during which stacking faults are removed. The Ar then accumulates into clusters leading to the creation of voids and blisters on the surface. Using molecular dynamics, the penetration threshold energies are determined for both Ar and Xe, with CdTe in both zinc-blende and wurtzite phases. These calculations show that more Ar than Xe can penetrate into the growing film with most penetration across the (111) surface. The mechanisms and energy barriers for interstitial Ar and Xe diffusion in zinc-blende are determined. Barriers are reduced near existing clusters, increasing the probability of capture-based cluster growth. Barriers in wurtzite are higher with non-Arrhenius behaviour observed. This provides an explanation for the increase in the size of voids observed after stacking fault removal. Blister exfoliation was also modelled, showing the formation of shallow craters with a raised rim.

## 1. Introduction

Thin film cadmium telluride (CdTe) is the most important thin-film photovoltaic technology with annual module

[†]Present address: Department of Electrical-Electronics Engineering, Duzce University, Duzce, Turkey.

**Figure 1.** A schematic diagram of a CdTe solar cell where TCO is the transparent conducting oxide layer and CdS is the *n-type* layer. (Online version in colour.)

production increasing to 8 GW. Manufacturing of CdTe solar modules is a comparatively simple and a relatively quick process resulting in low costs of production. The absorber layer is typically only 3 μm in thickness compared with 150 μm for crystalline silicon. This provides thin-film CdTe with a natural cost advantage. Module efficiency can now be greater than 19% and this exceeds the efficiency of polycrystalline silicon solar panels [1]. Currently, the champion conversion efficiency achieved from a CdTe solar cell in a research environment is 22.1% [2]. This has been achieved by adding selenium to the front of the cell and replacing the cadmium sulfide buffer layer with a transparent metal oxide. However, the theoretical maximum efficiency according to the Shockley–Queisser limit is 30% and this discrepancy is often attributed to the nature of the polycrystalline absorber [3].

CdTe crystallizes in a zinc-blende structure with a lattice constant of 6.48 Å [4]. During deposition, the CdTe film grows with the dominant surface in a (1 1 1) direction but if the surface is rough other crystal planes can be exposed [5]. It has been observed that a high density of stacking faults form, terminating at the grain boundaries. Twin boundaries are also often observed. So, while the film is mainly zinc-blende, wurtzite layers are also present [6].

CdTe is a *p-type* material with a direct band gap of 1.45 eV making it ideal for photovoltaic conversion. Its very high absorption coefficient also means that less than a 2 μm layer is required for total absorption of the solar energy spectrum [6]. A typical CdTe solar cell has superstrate configuration which is shown schematically in figure 1. A glass substrate is used which is coated with a transparent conducting oxide (TCO). The TCO acts as a front contact for the cell and is usually fluorine-doped tin oxide (FTO). In the traditional architecture, a thin buffer layer of *n-type* CdS is deposited followed by the *p-type* CdTe forming a *p-n* hetero-junction. The latest cells incorporate Se in the CdTe absorber to reduce the band gap and the CdS is replaced with magnesium-doped zinc oxide or some other transparent oxide layer. Both of these changes have improved the short circuit current [7,8]. A back contact with copper doping is then applied, which completes the cell [6].

The most common method currently employed in manufacturing to deposit CdTe films is vapour transport deposition (VTD) and the closely related close-space sublimation (CSS) method is often used to produce research devices [9]. Magnetron sputtering can also be used

with the advantage that this method deposits dense films with high uniformity [10,11]. An all-sputtered CdTe cell would lend itself to efficient high throughput, batch or in-line production. Recent adoption of pulsed DC sputtering has led to deposition rates of $3\,\mathrm{nm\,s}^{-1}$, high enough to be suitable for low-cost manufacturing [12]. The coating uniformity offered by sputtering would open up new markets such as power-producing semi-transparent coatings for residential windows or automotive glass applications.

The conversion efficiency of as-deposited CdTe cells is poor irrespective of the deposition technique used. It is always necessary to activate the cells with a cadmium chloride treatment typically at a temperature of $400°C$. The activation treatment has many effects including defect removal and front interface passivation. Thin films of as-deposited sputtered CdTe are columnar through the film. The grain size is small, typically less than 250 nm, and the activation process causes dramatic re-crystallization, re-orientation and growth of grain size. The initial grain size obtained using VTD and CSS is much larger and the effect of activation on the grain size is less pronounced.

However, a problem arises when CdTe is deposited by magnetron sputtering, which was first reported by Compaan & Bohn in 1994 using radio-frequency (RF) magnetron sputtering. It was found that after the $CdCl_2$ treatment, the CdTe surface is heavily blistered [13]. Recently, it has been discovered that the occurrence of these blisters is caused by the formation of argon gas bubbles which form during the high-temperature activation process [14]. The bubbles increase in size by agglomeration and this causes the surface blistering. The mechanisms leading to this gas bubble and blister formation have remained unexplained until now. Since the blistering problem is associated with inert gas retention in the films, this study compares the effects of using argon and xenon as the working gas. A combined experimental and modelling investigation presented here helps to explain both the processes involved and provide clues to how the sputter deposition processes might be improved to reduce or eliminate this problem. This is the purpose of this study.

## 2. Methodology

### (a) Pulsed DC magnetron sputtering

The devices were deposited using a 'PV Solar' sputtering system. Four circular magnetrons are mounted vertically around a vacuum chamber, with substrates also mounted vertically on a rotating carrier with a $\simeq10\,\mathrm{cm}$ gap between the target and the rotating substrate. Coating uniformity in the horizontal direction is achieved by rotation and uniformity in the vertical direction is achieved by trimming the sputtered flux with a suitably shaped disposable mask. Up to six 5 cm × 5 cm substrates are mounted on the carrier that rotates at $\simeq120\,\mathrm{r.p.m.}$. The argon working gas is admitted via a mass flow controller rated at 100 sccm with the pressure fixed to $\simeq3.3\,\mu\mathrm{bar}$. The films were sputtered using a 5 kW pulsed DC power supply (Advanced Energy Inc, Pinnacle Plus) with a pulsing frequency range from 150 to 350 kHz. The voltage on the target is typically $\simeq800\,\mathrm{V}$ during the sputtering process although this value depends on the resistivity of the individual target used. The system is equipped with radiant heaters and the substrate temperature during deposition can be maintained at temperatures up to $250°C$. This temperature and the use of relatively high working gas pressure with gas flows of 50 sccm has been found to minimize stress in the films [15].

The devices were deposited on NSG-Pilkington TEC10 glass coated with fluorine-doped tin oxide. The substrates were cleaned in a 10% iso-propyl alcohol in 18 Mohm-cm de-ionized water solution in an ultrasonic bath held at $60°C$ for 60 min. This was followed by a de-ionized water rinse and drying. The cadmium sulfide layer was then deposited by pulsed DC magnetron sputtering using process conditions described previously [16].

The CdTe films were then deposited from a semi-insulating, compound CdTe target by pulsed DC magnetron sputtering. Use of pulsed DC power results in much higher deposition rates than conventional RF sputtering due to the much improved duty cycle.

Cell activation is achieved using a CdCl$_2$ treatment. Cells are loaded into a vacuum evacuated tube furnace above $\simeq$0.5 g–1 g CdCl$_2$ powder. The system is sealed under a vacuum at 50–100 mbar. The device and the CdCl$_2$ are heated using infra-red (IR) lamps and kept at a temperature $\simeq$400°C and CdCl$_2$ vapour is formed. Chlorine diffuses into the device and activates it. The process takes $\simeq$30 min.

Transmission electron microscopy (TEM) was used to investigate the structure of the as deposited and activated CdTe devices. Samples for TEM were prepared by an *in situ* lift out method using an FEI Nova 600 Nanolab Focused Ion Beam (FIB), which was also equipped with an in-lens SEM detector used for detailed surface imaging. For preparing cross-sectional samples through the coating, a standard *in situ* lift out method was used. A layer of platinum (Pt) was deposited to determine the surface and homogenize the thinning of the films. TEM was performed using a Jeol JEM 2000FX, with an Oxford Instruments 30 mm$^2$ EDX detector and a Gatan Erlangshen ES500W digital camera. High-resolution (HR-TEM) images were obtained using an FEI Tecnai F20 scanning transmission electron microscope (STEM) to examine features in the device cross-sections with atomic resolution.

## (b) Modelling methodology

Molecular dynamics (MD) is the principal modelling tool used in the investigation, with force fields described by many-body potentials as used by previous authors [5,17]. The LAMMPS package was used for these calculations [18]. The Stillinger–Weber (SW) potential for CdTe was mainly used which predicts bulk-like properties well [19]. An alternative potential used for modelling CdTe is a bond-order potential (BOP), shown to model surface defects more accurately than SW [20] and used previously to model thin-film growth [21]. Simulations using the BOP potential are more computationally costly so only a sample of the results was verified using the BOP potential. The Ziegler–Biersack–Littmark purely repulsive screened Coulomb potential [22] is used to model the interaction of inert gases Ar and Xe with the CdTe layer. For Ar–Ar and Xe–Xe interactions, the Lennard–Jones potential parametrized in Ashcroft & Mermin [23] is used with a cut-off distance of 8.5 Å [24]. A small number of calculations were also carried out using the ParSplice technique [25].

MD was first used to determine the penetration energy thresholds, i.e. the lowest energy at which an incident inert gas atom can be implanted below the surface of CdTe. This was studied for three different zinc-blende surface orientations (1 0 0), (1 1 0) and (1 1 1) as well as analogous wurtzite surfaces, all of which were unreconstructed. For each orientation, we impacted atoms normally over an irreducible symmetry zone (ISZ), meaning that impacts onto this area would be representative of impacts over the whole surface. The simulations were run until the energy of the impacting inert gas atom dropped below 0.1 eV or had been reflected from the system. The value of 0.1 eV was chosen as it is substantially below the energy barrier for Ar diffusion in the lattice.

The lattice size was chosen as $5 \times 5 \times 10$ nm$^3$ and the results were carried out with the lattice initially at 0 K, this was done to obtain a small statistically valid data set by removing the random motion of atoms at a non-zero temperature. These simulations were then repeated at the experimental temperature of 523 K (250°C) to ascertain if temperature plays a significant role. In the latter case, the CdTe lattice was thermalized to 250°C using a Nosé–Hoover thermostat. Periodic boundaries were used except in the direction perpendicular to the surface where free boundary conditions were applied and an atom was positioned above the surface outside the range of the interaction potential with the substrate. We then gave the atom a specified energy and projected it normally towards a randomly distributed point on the ISZ and recorded if it bounced from the surface or implanted below the surface. In total, 861 impacts were analysed for each orientation and each deposition energy to obtain reasonable statistics.

In addition to the threshold energy and penetration depth, inert gas cluster growth was investigated, inert gas atoms were distributed randomly and interstitially throughout a larger system. A volume concentration of 4% was chosen for Ar to match the experimentally determined

value [26]. Periodic boundary conditions were applied in all directions which simulates a bulk structure without considering interactions with a free surface. This system was then annealed at temperatures of 700 K and also 1000 K for various lengths of time to investigate Ar clustering.

From the MD simulations, diffusion pathways could be identified and the corresponding energy barriers were determined using the nudged elastic band (NEB) method [27]. A few energy barriers were also determined by first principles techniques for comparison and by high-temperature MD simulations using the Arrhenius equation for the diffusion rate $R$,

$$R = A \, e^{-(E_a/kT)}, \tag{2.1}$$

where $A$ is the Arrhenius prefactor related to the rate of vibrational movements along the minimum energy path (MEP). $E_a$ is the activation energy, i.e. the energy barrier for the transition. $T$ is the temperature and $k$ is Boltzmann's constant [28]. To use this method, MD determines transitions times at various temperatures. It was later found that MD could only determine transitions at temperatures larger than $\approx$800 K, so to access lower temperatures, Parallel Trajectory Splicing (ParSplice) [25] was used for all temperatures. This allowed transitions to be found at temperatures between 1100 K and 600 K.

## 3. Results and discussion

### (a) Experimental results

A TEM image of a cross-section through an as-deposited CdTe film is shown in figure 2a. The image reveals that the CdS and CdTe layers are compact and void free. The coating is highly uniform and the surface roughness simply mimics the roughness of the original FTO-coated substrate. Measurements using X-ray diffraction show that the texture of the CdTe is highly (1 1 1) orientated [29]. The CdTe layer contains a high density of stacking faults which appear as parallel lines terminating at each end at grain boundaries, as seen in figure 3a. It has been estimated that there is a 48% incidence of the wurtzite phase in the as-deposited material [30]. Owing to a small energy difference in zinc-blende compared to wurtzite stacking, the layers of CdTe have an almost even probability of being a stacking fault where these stacking faults create at least one layer of wurtzite structured CdTe [31].

In the films grown by magnetron sputtering, elemental analysis of the as-deposited film using energy dispersive analysis (EDS) in the TEM reveals that argon is present at a concentration of about 4 At% [32]. The argon appears to be uniformly dispersed because there is no microstructural effect observed in the as-deposited device cross-section shown in figure 2a.

After the high-temperature CdCl$_2$ activation treatment, the stacking faults are removed [6] as shown in figure 3b and the remaining film is almost completely zinc-blende structured with a few twin boundaries remaining. Investigations into the mechanisms leading to stacking fault removal is an area of current research [30,31]. During this process, Ar agglomeration occurs and micrometre-sized bubbles form causing large internal voids, as shown in figure 2b. Some bubbles pierce the surface of the film as shown in figure 4. These can cause complete and catastrophic delamination of the CdTe from the CdS buffer layer and also result in surface blistering and surface exfoliation.

If the device is simply annealed at 350°C in a rapid thermal annealing (RTA) system, without the presence of cadmium chloride, the film structure and density of stacking faults remain the same but the argon diffuses to form small bubbles. The high-resolution TEM image shown in figure 5b shows the presence of several nanometre scale argon bubbles which straddle the stacking faults. These structures also appear to be located adjacent to grain boundaries or at the device junction.

Since Ar has an atomic mass much smaller than both Cd and Te, it is likely that this size difference allows for large amounts of penetration of Ar during sputtering as well increasing diffusion compared to a larger inert gas atom. Therefore, using a larger inert gas atom closer in mass to Cd and Te, such as Xenon, should reduce inert gas penetration and associated

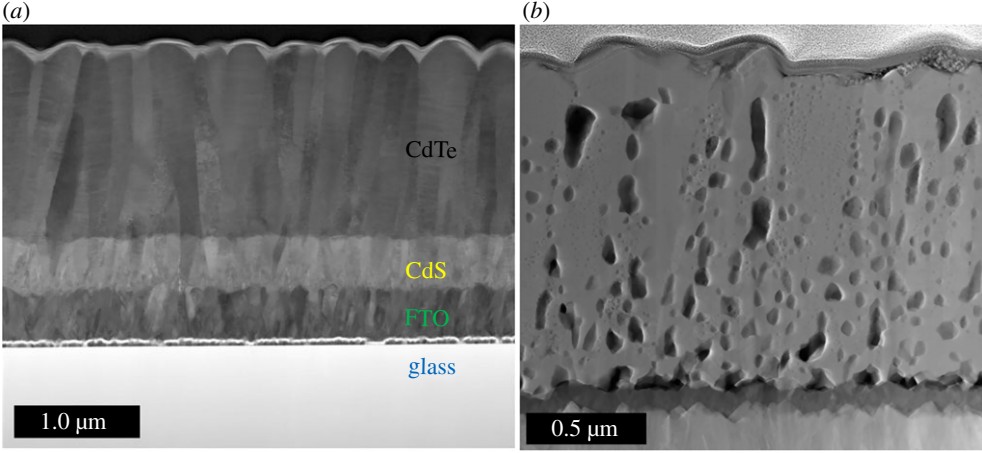

**Figure 2.** Low-resolution cross-sectional TEM images of a CdTe device (*a*) as-deposited by magnetron sputtering showing no microstructural effect of Ar and (*b*) after the high-temperature CdCl$_2$ treatment. The image in figure. (*b*) reveals the appearance of micron sized Ar bubbles or voids agglomerating below the surface of the cell. The voids accumulate along the CdS/CdTe interface and along grain boundaries. (Online version in colour.)

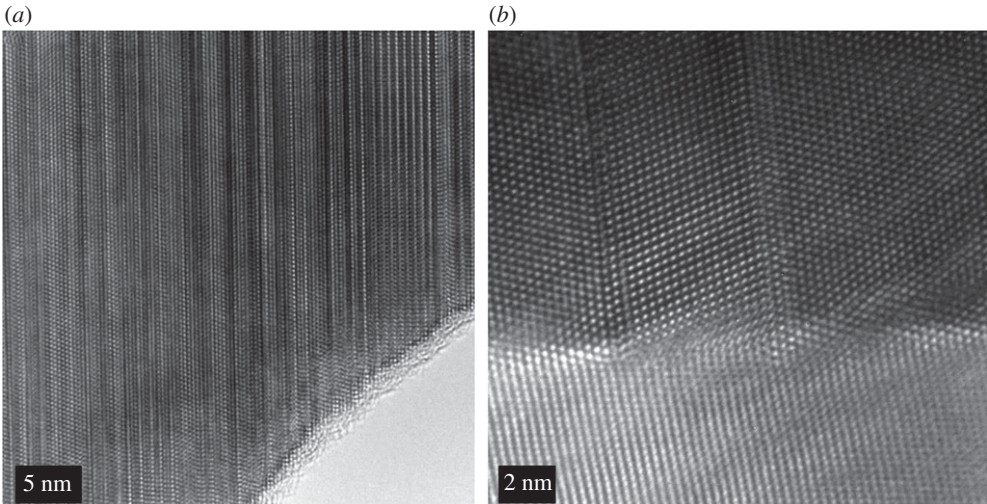

**Figure 3.** (*a*) A high-resolution TEM image of a cross-section of an as-deposited CdTe device deposited using magnetron sputtering showing a high density of stacking faults terminating at grain boundaries. (*b*) CdTe film deposited by magnetron sputtering after the CdCl$_2$ activation treatment. The stacking faults have been removed but twin boundaries remain.

implantation. It should also reduce clustering into bubbles and therefore blistering. Experiments with Xe have confirmed that a lower percentage (around 2%) is implanted, compared with Ar. Modelling the use of Ar and Xe will reveal the level of improvement expected if the magnetron working gas is switched from Argon to Xenon.

## (b) Molecular dynamics results

### (i) Deposition thresholds in zinc-blende CdTe

The source of the inert gas atoms embedded in the CdTe layer is because of the use of an unbalanced magnetron which results in energetic neutral working gas atoms assisting and

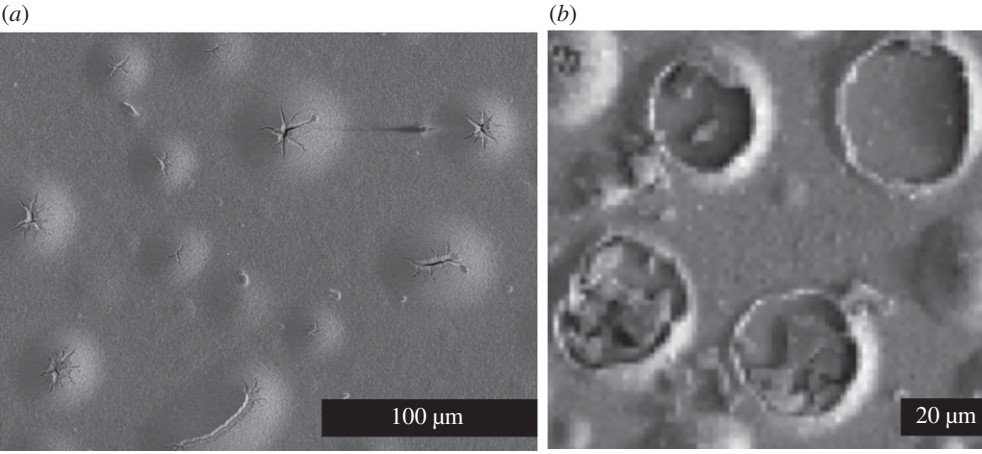

**Figure 4.** SEM images of blisters on the surface of sputter-deposited CdTe occurring after the CdCl$_2$ treatment; (*a*) showing almost intact surface structures; (*b*) examples where the blisters have 'exfoliated'.

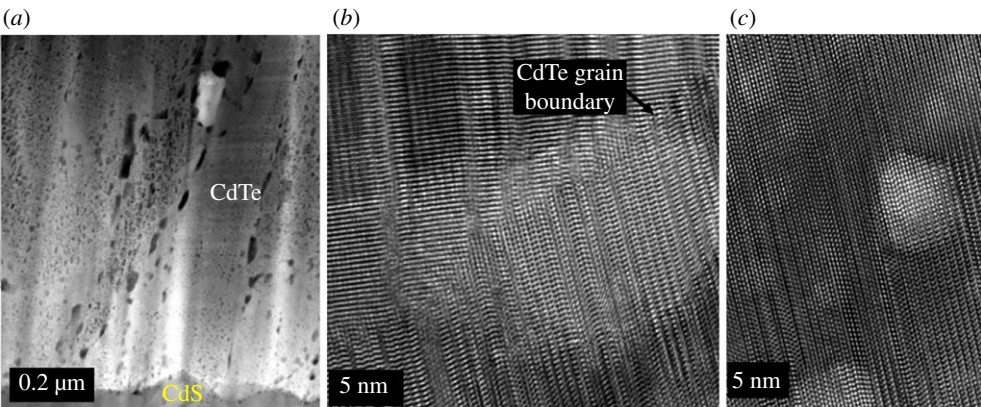

**Figure 5.** TEM images of the structure of a sputter-deposited film after thermal annealing for 12 h at 350°C. (*a*) Low-resolution image showing Ar clusters which are elongated structures which can be several tens of nanometres in length. (*c*) High-resolution image showing a 15 nm Ar bubble located against a CdTe grain boundary. (*c*) High-resolution image showing bubbles of Ar about 5 nm in diameter under the CdTe surface straddling the stacking faults. (Online version in colour.)

compacting the growing film. Therefore, we firstly seek to investigate the mechanisms and thresholds for this implantation.

Table 1 shows the results for the percentage of inert gas penetrating CdTe at different energies and through different zinc-blende surface orientations at 0 K. Calculations were also carried out at 523 K (250°C) but the only difference for the penetration threshold was a reduction from 9 to 8 eV for Xe(111) and from 15 to 10 eV for Ar(100). All other thresholds remained the same so only the complete table for the 0 K case is shown. As expected, an increase in the energy of an incoming atom increases the penetration probability and the larger atom Xe penetrates less than Ar. Also, penetration depends heavily on surface orientation, with (1 1 1) allowing for the most penetration of both Ar and Xe due to the open structure of its top layer compared with other orientations. The surface texture of sputtered CdTe is predominantly in the (1 1 1) orientation. Note also that the number of atoms that penetrate is a very small proportion of the 861 events analysed for each result in the table so a ±1% variation in the values shown is expected.

We have collated all the depth data from the three surface orientations into one table and organized the data in terms of which atomic layers they reach rather than by actual depth. The

**Table 1.** Table showing the amount of implanted Ar and Xe into zinc-blende CdTe for various surface orientations, at 0 K, as a function of deposition energy.

| | 5 eV | 6 eV | 7 eV | 8 eV | 9 eV | 10 eV | 15 eV | 20 eV |
|---|---|---|---|---|---|---|---|---|
| Ar(100) | 0% | — | — | — | — | 0% | 1% | 2% |
| Xe(100) | 0% | — | — | — | — | 0% | 0% | 1% |
| Ar(110) | 0% | — | 0% | 1% | 4% | 5% | 6% | 6% |
| Xe(110) | 0% | — | 0% | 0% | 0% | 0% | 4% | 4% |
| Ar(111) | 0% | 2% | 4% | 4% | 5% | 8% | 10% | 11% |
| Xe(111) | 0% | 0% | 0% | 0% | 4% | 7% | 10% | 11% |

**Table 2.** Table showing the penetration depths of Ar and Xe atoms at 0 K.

| | Layer 1 | Layer 2 | Layer 3 | Layer 4 |
|---|---|---|---|---|
| Ar 20 eV | 52% | 31% | 12% | 5% |
| Xe 20 eV | 59% | 30% | 5 % | 2% |
| Ar 15 eV | 62% | 21% | 15% | 2% |
| Xe 15 eV | 94% | 4% | 1% | 1% |
| Ar 10 eV | 100% | 0% | 0% | 0% |
| Xe 10 eV | 100% | 0% | 0% | 0% |

reason for this is that this method also defines the number of energy barriers an atom needs to overcome to escape from the surface. The data are shown in table 2. Data are only given for energies of 10 eV or greater since below that energy any atoms that implant do so in layer 1, i.e. the surface layer. In all cases the implanted atoms are interstitial and no Cd or Te atoms are displaced.

The energy barriers for Ar to escape from the surface are much lower than that for diffusion within the crystal and as will later be shown can be as low as 0.37 eV. The MD simulations run typically for around 10 ps. Such low diffusion barriers for atoms trapped just below the surface, would suggest that much of the gas implanted could escape before the next layer is deposited since typical experimental growth rates are typically $1 \, \mathrm{mLs}^{-1}$. Diffusion barrier calculations are discussed later, but assuming an escape barrier of 0.37 eV, a normal growth temperature of 523 K (250°C) and a typical prefactor of $10^{13} \, \mathrm{s}^{-1}$, the escape time would be around 20 ns. Thus, it might be expected that the results in table 1 for the degree of inert gas penetration near the threshold would be overestimated compared to experiment.

## (ii) Annealing simulations

In this section, the clustering of implanted Ar and Xe is investigated by performing high-temperature MD simulations. A typical starting configuration is shown in figure 6. The system was first relaxed, heated to 700 K and 1000 K and then held at those temperatures while being allowed to evolve for 3 ns. The final configurations of Ar are shown in figure 7. Over nanosecond time scales, Ar atoms have begun to cluster, the largest cluster at 1000 K contains 53 atoms with a diameter of around 3 nm.

At a lower temperature of 700 K, similar to that used in the $CdCl_2$ activation treatment, the number of large clusters is fewer but their structure and spatial distribution is equivalent to 500 ps of simulation time at 1000 K, so the 1000 K simulation forms clusters approximately six times faster than at 700 K. The most common cluster at 700 K contains two Ar with the maximum cluster size being 22.

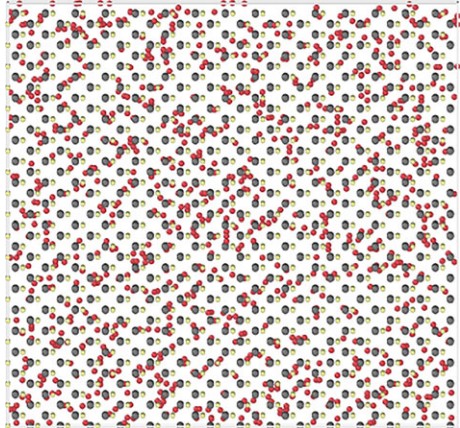

**Figure 6.** Plan view of the starting configuration for an annealing simulation of ≈20 000 atoms with Ar randomly distributed interstitially with a volume concentration of 4%, where red represents Ar, grey Te and yellow Cd atoms. (Online version in colour.)

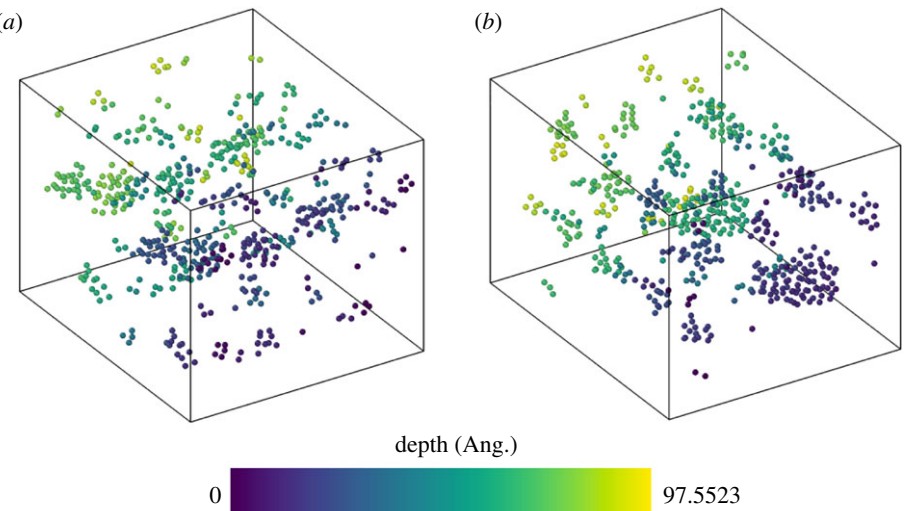

depth (Ang.)

0                                                              97.5523

**Figure 7.** The final Ar configuration of the (*a*) 700 K and (*b*) 1000 K simulations after 3 ns (colour coded based on depth) showing examples of nanometre sized clustering driven by a thermal process. The Cd and Te atoms are not shown for clarity. (Online version in colour.)

During annealing, single Ar atoms were observed to migrate from areas of a low concentration of Ar to areas of higher concentration. Usually the resulting clusters consisted of interstitial atoms but in a few cases there was evidence of a trap mutation mechanism [33] where a lattice atom is ejected as an interstitial and is replaced by a gas atom; this only occurs when a cluster reaches a size containing at least 6 Ar atoms. The consequence of this is that large clusters can become 'pinned' to their location since substitutional Ar atoms are more strongly bound. If Xe is distributed and annealed at 1000 K for 3 ns with 4% concentration, clusters develop similarly to those with Ar. Calculations show that 3 ns annealing time at 1000 K for Xe produces a cluster size distribution equivalent to ≈750 ps in the case of Ar. Although the cluster aggregation rate for Xe is slower, the results indicate that Xe inert gas clusters would also form in the CdTe film over experimental time scales if sufficient Xe penetrates into the surface layers. Free surface calculations have also been conducted in which one of surfaces' periodic boundary conditions

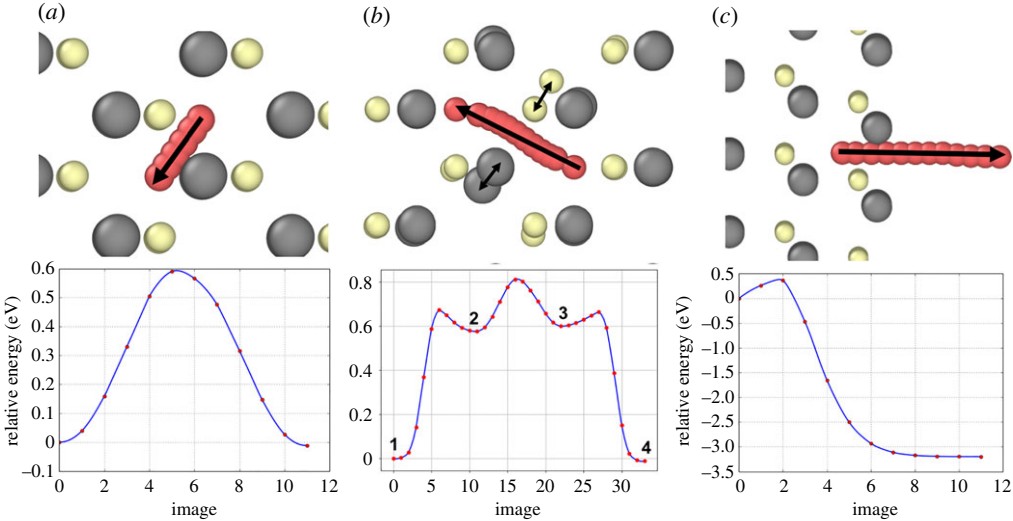

**Figure 8.** Various diffusion mechanisms for a single Ar interstitial. (*a*) Direct transformation from a Cd-coordinated interstitial site to a Te-coordinated site with an energy barrier of 0.6 eV shown onto the (110) plane; (*b*) A 3-stage mechanism through two dumbbell type positions shown on the (110) plane. Minima (1) and (4) are identical in structure to the start and end of transition (a), respectively. Minimum (2) is a dumbell structure in which Ar is sharing a lattice site with the displaced Te atom (grey). Minimum (3) is also a dumbell structure with the same Te but the Ar has moved to the adjacent symmetrical site. Finally, the dumbell is released and Ar relaxes to minimum (4); (*c*) escape from the (1̄11̄) surface with a 0.37 eV energy barrier. (Online version in colour.)

were removed and atoms could escape the CdTe. Upon heating in cases described previously, we found that only Ar/Xe atoms in the top two layers were able to escape, below that the effect of the free surface was not felt and diffusion was mainly into clusters as discussed above.

### (iii) Energy barriers for diffusion in zinc-blende CdTe

The principal diffusion mechanism for a single Ar atom is shown in figure 8*a* with an energy barrier of 0.6 eV. The barriers were determined by the NEB method and the energy profiles along the minimum energy pathways are also shown. The start and end sites demonstrate the two types of tetrahedral stable sites, Cd-coordinated and Te-coordinated, respectively, which are not symmetric but almost equivalent in energy.

Figure 8*b* shows another mechanism, whereby the Ar atoms form a metastable dumbbell type configuration with a Te atom and from there, two lower energy barrier hops occur, first allowing the Ar to move into an adjacent dumbell site and then to the adjacent stable interstitial site. The escape barrier from the (1 1 1) surface is also shown in (c) with an energy barrier of 0.37 eV.

The transition times for a single Ar atom to diffuse can be determined directly from MD. These transition times were then averaged over 10 different events at each temperature. By using equation (2.1) and performing a log plot of $R$ against $1/T$ we can also estimate both the energy barrier and the prefactor, $A$. Figure 9 shows that $A \approx 4 \times 10^{13}\,\text{s}^{-1}$ and $E = 0.67\,\text{eV}$. This is slightly higher than the energy barrier associated with the most favourable pathway indicating that occasionally MD found the alternative transition pathway shown in figure 8*b* with the higher barrier. Using this prefactor and the diffusion barriers, it is possible to estimate the time taken per single hop at different temperatures. Table 3 shows the time taken for a single hop through the mechanism described in figure 8*a*.

Similar calculations for Xe show identical pathways are possible as for Ar but with higher energy barriers. The barriers for Xe diffusion are 0.74 eV, 0.9 eV and 0.52 eV for the pathways described in figure 8, respectively.

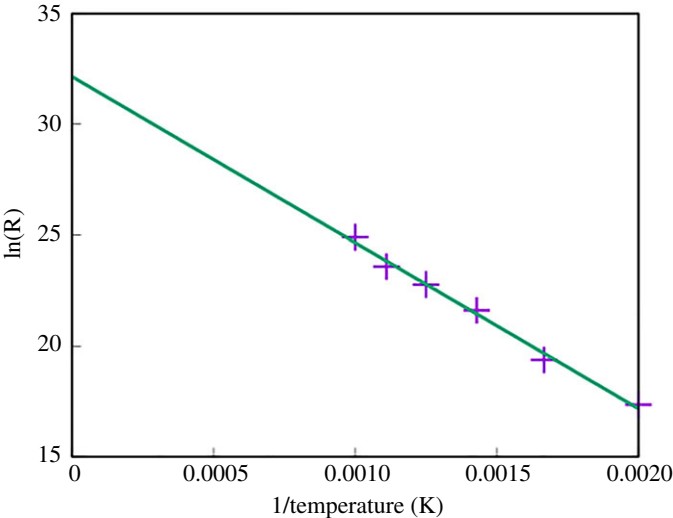

**Figure 9.** Arrhenius plot of the motion of a single Ar atom in a perfect CdTe lattice. The gradient indicates an overall energy barrier of 0.67 eV and the estimated prefactor for use in the Arrhenius equation (2.1) is $\approx 4 \times 10^{13}\,s^{-1}$. (Online version in colour.)

**Table 3.** A table showing the approximate hop times for an Ar atom to diffuse in a perfect CdTe lattice at different temperatures.

| temperature (K) | approximate hop time (ns) |
|---|---|
| 300 | 310 000 |
| 500 | 28 |
| 700 | 0.5 |
| 1000 | 0.03 |

The annealing simulations show that a large number of di-interstitial clusters form during the clustering process. This is due to a low barrier of 0.2 eV for a di-interstitial to form providing two Ar/Xe atoms are in adjacent interstitial sites. The elevated reverse barrier of 0.9 eV for the dissipation of the di-interstitial also contributes to the large number of these defects [34]. Moreover, energy barriers and pathways for di-interstitial Ar diffusion were also determined. The di-interstitials can rotate (but not diffuse) with low-energy barriers of less than 0.04 eV. The di-interstitial can move with a two-stage catch-up mechanism by which it first splits to a metastable site with a barrier of 1 eV followed by one of the atoms breaking away with a barrier of 0.5 eV before the second atom rejoins.

So far, energy barriers have been examined for atoms diffusing through a perfect CdTe lattice. However, analysis using EDS has shown that 4% Argon is incorporated into the CdTe layer. Therefore, a distribution of barriers is determined when 4% is already present in the film. Figure 10 shows a box plot using data from NEB calculations of Ar diffusing in three different systems from a random distribution of sites. For system (1), the two energy barriers shown earlier are given. In (2), the mean energy barrier reduces to 0.52 eV but the minimum barrier is 0.3 eV. A similar reduction is seen in scenario (3) and as a result of the lowered energy barriers, diffusion will be faster. For context, using energy barriers given in figure 10, equation (2.1) and the Arrhenius prefactor seen in figure 9, we present table 4 which shows approximate hop times for some representative energy barriers at 700 K.

The lowest diffusion barriers shown in figure 10 were obtained in scenarios where Ar moved from an area of low Ar concentration to one of higher Ar concentration. Figure 11 illustrates the

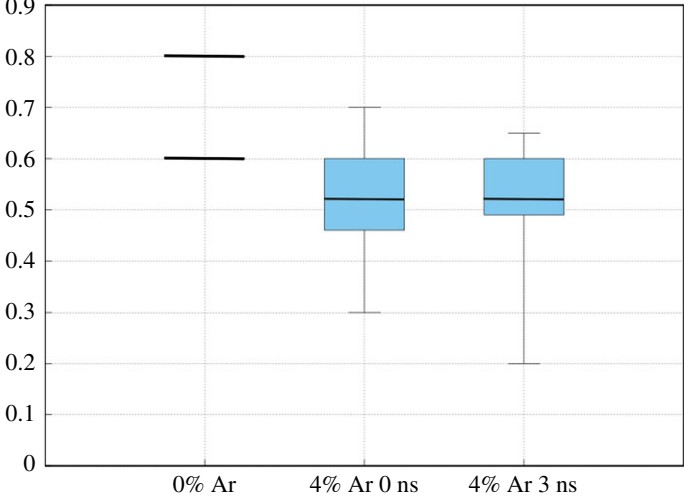

**Figure 10.** Energy barriers in three different scenarios of argon in the zinc-blende CdTe. (1) 0% Ar: (2) 4% Ar distributed randomly throughout. (3) 4% 3 ns corresponding to the distribution shown in figure 7*b*. The blue shaded areas represent the upper and lower quartile regions of the barrier distributions in (2) and (3), with the bold horizontal line being the mean; the vertical lines represent the spread of values. (Online version in colour.)

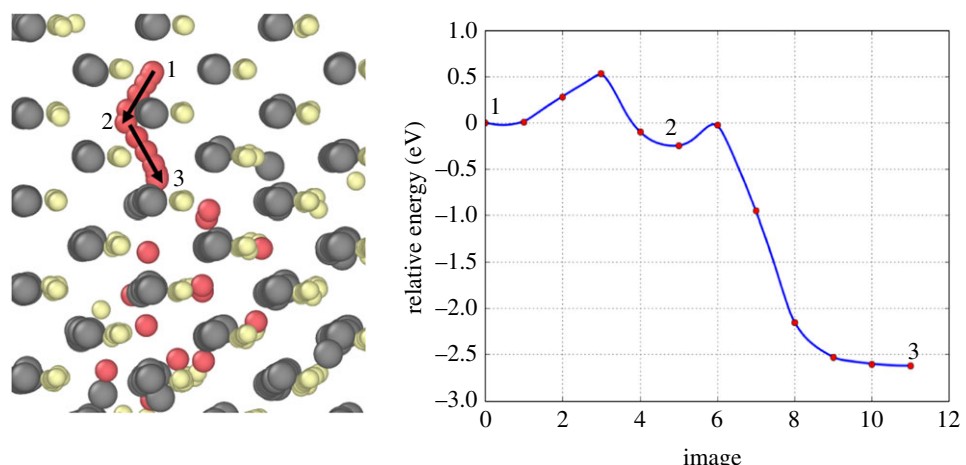

**Figure 11.** The path and potential energy landscape of a single Ar atom diffusing into a 13 atom Ar cluster. The positions of the Ar atoms (red) correspond sequentially to the points on the energy landscape. (Online version in colour.)

**Table 4.** A table showing the approximate hop times for an Ar atom to diffuse in a perfect CdTe lattice at different temperatures.

| energy barrier (eV) | approximate hop time (ns) |
| --- | --- |
| 0.2 | $7.0 \times 10^{-4}$ |
| 0.3 | $3.6 \times 10^{-3}$ |
| 0.52 | 0.14 |
| 0.6 | 0.53 |
| 0.8 | 15 |
| 1.0 | 410 |
| 1.2 | 11000 |

**Table 5.** The percentage of Ar and Xe at 0 K implanted into wurtzite CdTe as a function of impact energy.

| | 5 eV | 6 eV | 7 eV | 8 eV | 9 eV | 10 eV | 15 eV | 20 eV |
|---|---|---|---|---|---|---|---|---|
| Argon - (111) | 0% | 3% | 7% | 10% | 12% | 14% | 18% | 20% |
| Xenon - (111) | 0% | 0% | 0% | 1% | 3% | 3% | 4% | 6% |

kind of relative energy landscape typical for this kind of transition. Position 1 relates to an Ar atom located more than two nearest-neighbour positions from a cluster of 13 Argon atoms and position 3 represents the final position of that Ar atom having joined the 13 atom cluster. The first barrier, (1) to (2), is similar in height to those seen in bulk transitions ($\approx 0.52$ eV) but the second barrier, (2) to (3), is lower at $\approx 0.25$ eV. There is also a large decrease in relative energy, $\approx -2.5$ eV after the Ar joins the cluster. This decrease occurs because the lattice strain is greater for the isolated Ar atom and the 13 atom cluster compared to a single 14 atom cluster. This promotes cluster growth by single-atom migration and restricts the possibility of the cluster dissipating since the high reverse barrier would need to be overcome.

In zinc-blende CdTe, figure 11 illustrates the principle of the main mechanism for Ar/Xe cluster growth. Clusters 'attract' single Ar/Xe atoms by creating a lattice distortion which allows for faster diffusion into the cluster. Once having joined, diffusion away is limited due to the high reverse barrier. Clusters are free to grow and further investigation has not found a cluster size which inhibits the mechanisms for cluster growth. Moreover, even clusters of 53 atoms encourage Ar/Xe incorporation through reduced barriers for faster diffusion into the cluster and energy decreases once the atom has joined. Therefore, we have found no evidence for a maximum cluster size in bulk zinc-blende CdTe.

We have demonstrated above that the dissipation of Ar clusters will be rare due to large barriers for atoms to leave a cluster. The binding energy of Ar/Xe atoms to their respective clusters will help indicate in general how likely clusters of different sizes are to disassemble. In small clusters of size 2–5 atoms, the binding energy of a single atom to the cluster is between $\simeq -1.1$ eV and $\simeq -1.3$ eV for Ar and Xe. For larger clusters of 10 atoms, the binding energies are $\simeq -1.8$ eV for both Ar and Xe. For larger clusters of around 30 atoms in size, the binding energy is $\simeq -2.9$ eV for both Ar and Xe. These binding energies are relatively large when compared with diffusion barriers of these atoms in a bulk system and are therefore not likely to be overcome under experimental temperatures which will restrict the likelihood of dissipation of clusters even as small as five atoms.

### (iv) Wurtzite CdTe

Since sputtered layers of CdTe contain a high density of stacking faults, locally within an as-deposited structure, layers of wurtzite are present [6,31]. Thus, MD calculations are also carried out for the wurtzite structure.

Table 5 is analogous to table 1 for zinc-blende. The degree of penetration is slightly higher than in zinc-blende. The penetration depths are also slightly deeper. The mechanisms for Ar and Xe diffusion in wurtzite are now investigated.

All stable sites for a single Ar/Xe atom in wurtzite are symmetric and two non-equivalent pathways exist for diffusion. These are shown in figures 12 and 13. The Ar transitions have energy barriers of 1.56 eV and 1.15 eV, respectively. Transitions were also examined in MD and ParSplice. Parsplice was required to access the lower temperatures since the transition times could not be accessed in a reasonable computational time with MD. Figure 14 shows the log plot of the diffusion times of these transitions against $1/T$. Non-Arrhenius behaviour is observed and at temperatures between 800 K and 1000 K an activation energy ($E_a$) of 0.64 eV is implied. However, at lower temperatures (less than or equal to 800 K) the activation energy is 1.4 eV in agreement with the NEB calculations. Results for Xe show a similar non-Arrhenius trend with the two barriers equivalent to barriers of 0.7 eV above 800 K and 1.5 eV below.

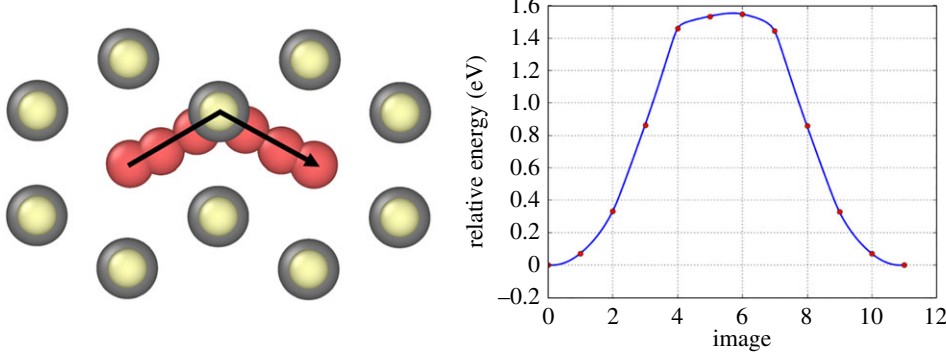

**Figure 12.** Ar/Xe transition in CdTe wurtzite. The energy barrier is 1.56 eV for Ar and 1.92 eV for Xe. (Online version in colour.)

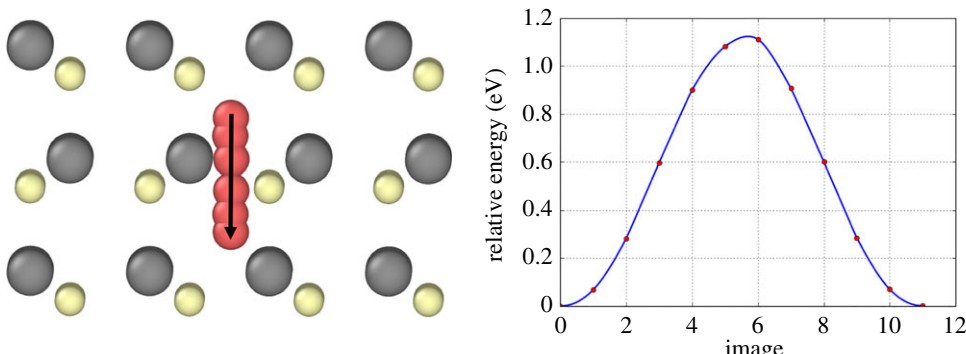

**Figure 13.** Ar/Xe transition in CdTe wurtzite. The energy barrier is 1.15 eV for Ar and 1.41 eV for Xe. (Online version in colour.)

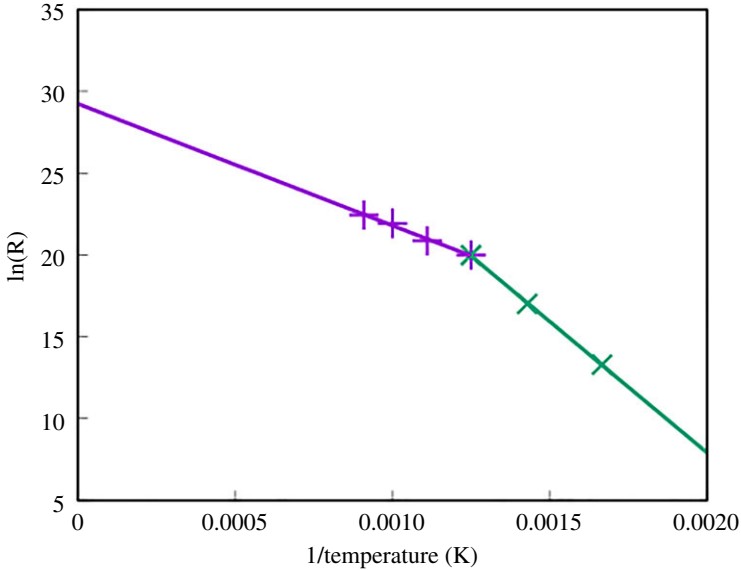

**Figure 14.** Log plot of the transition rate of a single Ar atom in a perfect wurtzite CdTe lattice showing non-Arrhenius behaviour. Assuming a piecewise linear curve, at temperatures >800 K the gradient would give a diffusion barrier of 0.64 eV and a prefactor of $4.8 \times 10^{12}$ $s^{-1}$. At temperatures less than 800 K the gradient indicates an energy barrier of 1.4 eV. (Online version in colour.)

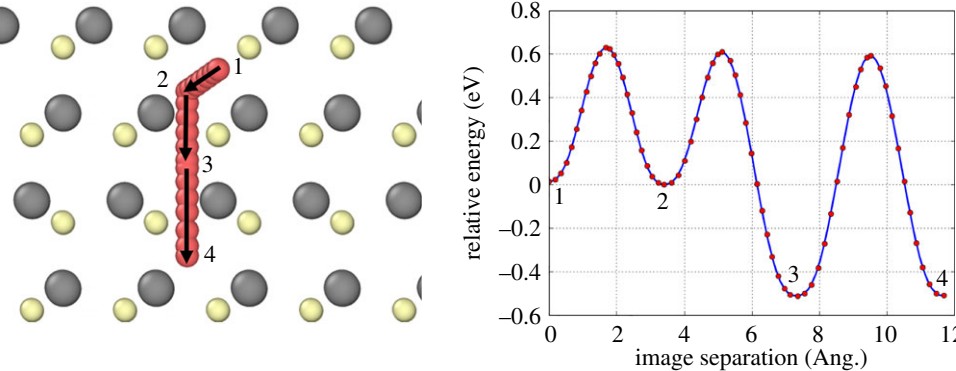

**Figure 15.** NEB calculation of Ar diffusion from a zinc-blende structure (Ar positions 1 and 2), formed by the top two layers of CdTe, into a wurtzite structure (Ar positions 3 and 4) which is induced by the stacking fault in layer 3. A large energy decrease is associated to the transition of Ar from zinc-blende into wurtzite due to the open structure of wurtzite. This results in Ar requiring a ~1.2 eV barrier for either diffusion which means the Ar is effectively trapped at the stacking fault. (Online version in colour.)

This non-Arrhenius behaviour is a rare finding, especially in solid-state modelling, where large structural changes are not occurring. As a result of this behaviour, any form of traditional temperature-assisted dynamics where the user crosses 800 K would result in findings that would not be representative of results at the original temperature. In many systems, increasing the temperature to simulate transitions faster is a valid method as it is assumed, sometimes correctly, that the Arrhenius plot is linear as in figure 9, however, as we have shown here it is important to check.

Since the temperature used in the $CdCl_2$ treatment is $\approx 400°C/673$ K, below the 800 K threshold, Ar and Xe diffusion in wurtzite is assumed to be slow compared to zinc-blende. We therefore conclude that no clustering will occur in wurtzite-structured layers as a single-energy barrier of 1.4 eV would take on average 0.01 s at 400°C.

The energy landscape for Ar diffusing across the zinc-blende/wurtzite interface is shown in figure 15. The site at the exact interface (Image separation $\simeq 7.5$ Å) has relative energy $\simeq 0.55$ eV below the stable site energy in zinc-blende (Image separation $\simeq 3.5$ Å). The figure also shows that the relative saddle point energy for Ar diffusion in wurtzite is also the same as the relative saddle point energy in zinc-blende ($\simeq 0.6$ eV).

Comparing the Ar interstitial in zinc-blende and wurtzite, it is found that the wurtzite site is more stable with a difference of $\simeq 0.55$ eV in the binding energy. This is due to the more open lattice around the Ar site in wurtzite meaning the lattice distortion caused by Ar is less than in zinc-blende structure. The diffusion barrier of an Ar atom in wurtzite and that to escape from across the zinc-blende/wurtzite interface is double that of diffusion in zinc-blende we therefore conclude this interface can act as a 'trap' for the diffusing Ar.

In order to have some confidence that the energy barriers calculated using the empirical potentials are a reasonable approximation, calculations were also carried out for interstitial Ar diffusion using density functional theory (DFT) in small systems containing $\approx 64$ atoms with the VASP code [35]. In zinc-blende, DFT could distinguish an energy difference between the beginning and end sites which were shown in figure 8a. The forward barrier was estimated as 0.4 eV and the reverse barrier as 0.8 eV, both similar to our estimate of 0.6 eV. In wurtzite, the two sites shown in figure 12 have the same energy but the barrier increased to 2.3 eV suggesting even less diffusion in the wurtzite structure compared to zinc-blende than predicted by the empirical potentials.

Our model suggests therefore that Ar clusters would not grow across stacking faults. This is confirmed by figure 5c where small clusters appear pinned between stacking faults. Figure 5a also supports this hypothesis. Annealing without the cadmium chloride treatment does not remove

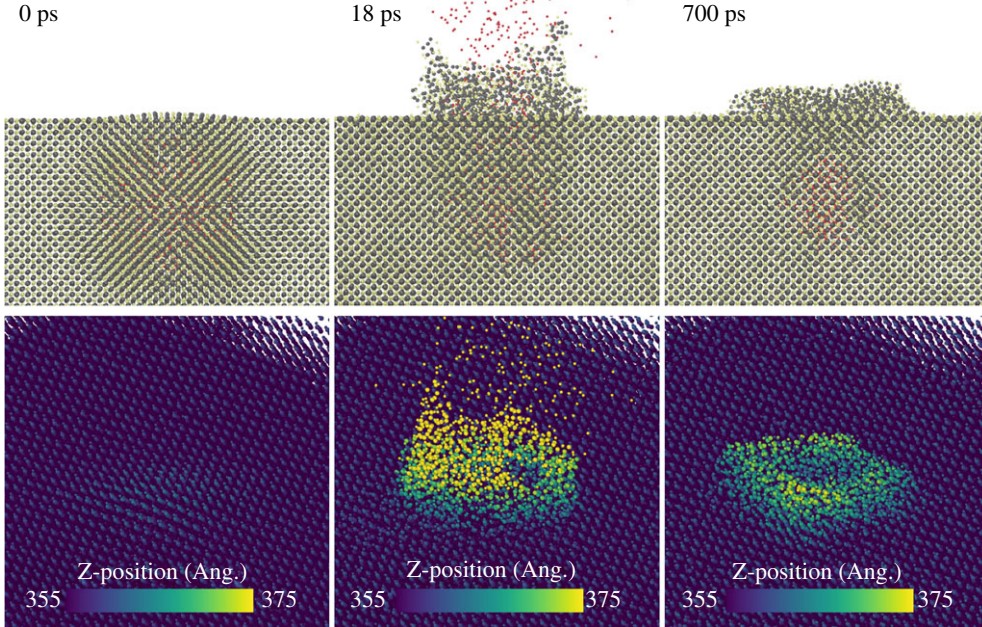

**Figure 16.** MD simulations of various stages in the bursting of a 6 nm diameter Ar bubble located 1 nm below the (111) surface of zinc-blende CdTe. The resulting surface structure closely resembles the exfoliation images such as that shown in figure 4b. (Online version in colour.)

stacking faults so we would expect the Ar clusters to grow non-spherically with the stacking faults acting as a barrier. Such elongated clusters are clearly visible in figure 5a. Once, the cadmium chloride treatment has removed the stacking faults, the clusters are free to grow since the large barriers for diffusion disappear and this leads to the micrometre-sized blisters seen in figures 2b and 4.

### (v) Surface rupture and blister exfoliation

In order to model the blister exfoliation, a large MD simulation containing 1.5 million CdTe atoms was set up with a free (1 1 1) surface with periodic boundary conditions in directions orthogonal to the surface and a fixed bottom layer. Thousand Ar atoms were inserted at various depths below the surface in an approximate spherical shape of diameter 6 nm. This is much smaller than the blisters observed experimentally but should give some insight into the mechanisms by which trapped Ar below the surface can escape.

The system temperature was set to 700 K and then the system was left to evolve for up to 1 ns; such simulations require large computing resources. Thus, a full systematic study of the precise conditions under which the blisters emit the trapped gas was not undertaken. When the top of the sphere was set to 1 nm below the surface the system became unstable and the bubble exploded as show in figure 16. The Ar atoms are drawn as red spheres in the figure.

Initially (0 ps), the main atom displacement is along the ⟨1 1 1⟩ directions. This causes the surface to bow upwards as well as atom displacements along equivalent directions as shown in the 0 ps image. After the initial explosion, the surface reforms after 0.7 ns. There is a crater rim about 2 nm in height around the edge of the hole created by the explosion and also the CdTe has formed a cap above the Ar that remains trapped below the surface. About 40% of Ar remains at a depth of 3 nm or more after the release. The SEM images of the blisters shown in figure 4b also show rims around what appear to be fairly shallow craters in agreement with the simulation results.

# 4. Discussion and conclusion

Thin-film cadmium telluride is the lowest cost solar technology; electricity from large-scale CdTe installations is being sold at less than $0.03 per kWhr. Most commercial CdTe modules are manufactured using vapour transport deposition as the tool to deposit the absorber layer. This technique produces CdTe with significant surface roughness. Magnetron sputtering is a deposition technique commonly used in industry to deposit coatings for a range of applications including multilayer optical coatings where coating uniformity is vital. However, depositing CdTe by magnetron sputtering at high deposition rates results in the problem of internal voids and surface blistering. In this paper, we have uncovered the mechanisms that cause this unusual problem.

Depositing CdTe by magnetron sputtering results in the magnetron working gas being implanted in the growing film. Using EDS analysis, we have measured that $\simeq 4\%$ Ar is implanted in the as-deposited CdTe film. As-deposited CdTe films contain a high density of stacking faults terminating at grain boundaries. High-resolution TEM images obtained from device cross-sections following annealing without the presence of cadmium chloride have shown that the Ar diffuses and is arranged in interstitial clusters about 5 nm in diameter. These clusters (or bubbles) are located near stacking faults, grain boundaries or at the *p-n* junction. Thin-film CdTe devices require activation using temperatures of 400°C in the presence of cadmium chloride. This activation treatment removes the high density of stacking faults present in as-deposited CdTe. We have shown that the stacking faults pin the movement of inert gas atoms but once removed the Ar diffuses rapidly to form large micrometre-sized bubbles which cause voids in the device cross-section and huge blisters on the surface. The formation of argon bubbles, voids and blisters is highly detrimental to the photovoltaic device performance. It is clear that the blistering is caused by significant implantation of Ar during the deposition process. If the magnetron working gas is switched to Xe, then the degree of implantation is reduced (2 At%) and this should reduce the problem. For this reason, we have compared the behaviour of Ar and Xe.

As-deposited CdTe takes the zinc-blende form with a surface in the (1 1 1) orientation. Using MD, it has been shown that working gas inclusion begins with incident energy 6 eV for Ar and 9 eV for Xe in zinc-blende (1 1 1) and 6 eV for Ar and 8 eV for Xe in wurtzite. It is unfortunate that the texture of sputtered CdTe is predominantly (1 1 1) orientated since this presents the most open structure to the incident inert gas flux and results in the highest level of implantation.

It has been shown that isolated inert gas atoms will diffuse in zinc-blende with overall barriers of 0.67 eV and 0.8 eV, for Ar and Xe, respectively. Both gases diffuse with the same two transitions which contribute to these overall barriers. One is a simple pathway through the most open lattice structure between two stable sites and the other is a more complex multi-stage transition through two intermediate dumbell-type interstitial positions. The transitions barriers for diffusion are reduced substantially in the presence of other inert gas atoms, reducing to as low as 0.2 eV in some cases. There is also a substantial drop in the system energy when single atoms join clusters. This is due to the relaxation of lattice atoms during this process, which means that clusters can form readily and dissipation is rare due to a high reverse barrier caused by the drop in system energy. This means clusters can form over MD time scales.

In wurtzite, isolated Ar and Xe atoms diffuse with overall barriers of 1.4 eV and 1.5 eV, respectively. This means that clusters will not grow as readily in wurtzite. Both gases diffuse with the same two transitions which contribute to these overall barriers. The first is a one-dimensional transition parallel to the (1 1 1) direction and the other is a two-dimensional transition perpendicular to the (1 1 1) direction. The Arrhenius plot reveals non-Arrhenius behaviour where the diffusion barrier depends on the temperature of the system meaning reliable non-trivial temperature assisted dynamics cannot be completed.

The barrier for an Ar atom to transition to a wurtzite region from a zinc-blende region (stacking fault) is 0.6 eV but the reverse barrier is $\approx 1.2$ eV due to a 0.6 eV more stable site in a wurtzite structure. Thus inert gas atoms will become 'trapped' at the interface of the wurtzite/zinc-blende stacking fault structure. This is confirmed by the high-resolution TEM images which shows

small clusters attached either to stacking faults, grain boundaries or to the CdS interface and after annealing, elongated clusters along grain boundaries implying a barrier for the inert gas to cross. It is the conclusion of this work that bubble growth is thermally driven, but the large proportion of stacking faults in the pre-treated film restricts the bubble size during a purely thermal process, such as the previously described thermal annealing in figure 5. We believe that the ability of the $CdCl_2$ to remove the majority of stacking faults enhances bubble growth by removing the wurtzite-structured barriers for bubble size. This means atoms can diffuse more quickly with energy barriers associated with a zinc blende structure, much lower than the analogous transitions in wurtzite. Therefore, micrometre-sized bubbles are free to grow under these conditions.

If sputter-deposited thin films of CdTe are to be used for solar applications, it is essential that the inclusion of inert gas is minimized since it leads to the formation of voids and surface blistering. In addition, it has also been observed that delamination at the CdTe/CdS interface can occur due to large-scale agglommeration there of the inert gas [34].

The MD results presented here suggest that sputtering with Xe could be a way in which inert gas clustering is reduced since the penetration barrier into the surface is higher than that for Ar. However, if Xe is implanted in the CdTe lattice it will cluster in a similar way as Ar.

The degree of inert gas implantation is dependent on the inert gas energy, and this is determined by a number of factors. The gas will be ionized close to the magnetron target and its energy will be governed by the potential difference between target and substrate. The ionized inert gas will neutralize and slow with collisions with other working gas atoms on its journey to the substrate. Hence the energy can be controlled and reduced by increasing the working gas pressure or by applying a DC bias to the substrate. Biasing the substrate in RF sputtering has resulted in efficient devices at low deposition rates but void formation has still been reported [36,37]. By manipulating these factors and by using Xe as the working gas, it should be possible to produce blister free devices at high deposition rates using pulsed DC power.

Magnetron sputtering is a widely used and industrially capable thin-film deposition technique routinely used by glass manufacturers for large area coating. The technique produces dense and uniform coatings. Its use for the deposition of thin-film CdTe solar cells has been held back because the absorber suffers from voids and blistering on the CdTe surface. This paper has revealed the mechanisms that lead to the formation of inert gas bubbles that are responsible for these problems. The key is to reduce inert gas implantation by reducing the energy of the ion-assist. This can be achieved by increasing the working gas pressure or by biasing the substrate. Ion implantation is also reduced by switching the working gas from argon to xenon. Using these modifications should allow the use of high rate pulsed DC sputtering with the promise of an industrially viable process for the deposition of uniform and stable thin film cadmium telluride devices.

Data accessibility. The underlying data for this article are available at: https://figshare.com/s/27bef54a3e412 ebb3d6e.

Authors' contributions. All authors contributed to the writing and revision of the manuscript.

Competing interests. We declare we have no competing interests.

Funding. The authors are grateful for funding through the EPSRC Supergen SuperSolar Hub (grant no. EP/J017361/1).

Acknowledgements. The authors acknowledge the use of Athena at HPC Midlands+, which was funded by the EPSRC on grant (EP/P020232/1), in this research, as part of the HPC Midlands+ consortium. The authors also recognize the use of the 'Hydra' High Performance System at Loughborough University.

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
