## [Reviewer comments · Proceedings. Mathematical, Physical, and Engineering Sciences]

Review History

RSPA-2020-0056.R0 (Original submission)

Review form: Referee 1

Is the manuscript an original and important contribution to its field?

Excellent

Is the paper of sufficient general interest?

Good

Is the overall quality of the paper suitable?

Excellent

Can the paper be shortened without overall detriment to the main message?

Yes

Do you think some of the material would be more appropriate as an electronic appendix?

No

Do you have any ethical concerns with this paper?

No

Recommendation?

Accept with minor revision (please list in comments)

Comments to the Author(s)

The authors present an interesting work combining experimental investigations with simulations for a better understanding of the formation of Ar clusters and blisters during sputtering of CdTe films.

Minor changes and additional explanations are necessary prior to publication.

- (1) In Figs. 2 - 5 electron microscopy graphs are shown. From my understanding there are TEM and also SEM images presented. Please specify in the Figure captions and add also the methods in Section 2 Methodology in more detail.
- (2) The existence of the Ar clusters is based on the electron microscopy investigations. Especially TEM preparation with FIB or other techniques can result in contamination and artifacts with such very delicate gas structures. Did you check the effect of preparation on the formation of the Ar clusters? Are there analytical methods (other than microscopy) to prove the formation and existence of the Ar clusters?
- (3) On p. 7 it is reported that 4% Ar is detected by EDS in the as-deposited films uniformly distributed over the whole CdTe layer of about 2-4 μm . However, if you compare with Fig. 2b the concentration of the Ar agglomeration in large internal voids and blisters seems to be higher than 4%. I guess Fig. 2b represents the cross section. Please give an explanation. A similar effect is also visible in Fig. 5 with blisters on the surface. The concentration seems to be larger than 4%. Can the Ar concentration be estimated directly from the images ?
- (4) I also miss a quantitative analysis of the Ar clusters in size and size distribution from the experiment. The simulation model predicts interstitial type Ar clusters with larger ones pinned by replacement of lattice atoms by Ar. Both types are correlated with stress fields in the CdTe film. Have you measured stress in the film, and can this be predicted from the simulations maybe also with a variation of the Ar content?
- (5) In the simulation model, it is reported that the most common Ar clusters after short time annealing are di-interstitials in the zinc-blende structure. Is there any explanation how they can form so favourably at elevated temperatures and how they are captured in the CdTe lattice structure?
- (6) The formation of larger Ar clusters in the zinc-blende structure is correlated with the displacement of a lattice atom by an Ar atom when the cluster contains 6 or more Ar atoms. Then the Ar cluster is pinned. What would be the effect of a free surface in the simulation. Would you expect most of the Ar to diffuse out of the lattice in the case of a free surface in a purely zinc-blende system. Maybe this is why sputtered CdS exhibits no bubbles and it is only the pinning of Ar by the stacking faults that causes Ar clusters in CdTe. It would be interesting to undertake a simulation with a surface, with and without a stacking fault near the surface.
- (7) In the wurtzite structure the energy barriers for diffusion are higher than in the zinc-blende structure and also 2 activation energies are detected (Fig. 14). For temperatures above 700 K the diffusion barrier is only 0.64 eV compared to 1.4 eV below 700 K. It is discussed that the Ar in a wurtzite state is more stable than in the zinc-blende structure since the lattice distortion by Ar is less. How can this be correlated with the 2 activation energies observed in wurtzite and the much lower activation energy at elevated temperatures?
- (8) How big are Ar clusters in the wurtzite structure? It seems that the Ar atoms are less mobile at least at low temperature. Is the low mobility related only to interstitial Ar clusters or is there also displacement of lattice atoms by Ar atoms observed?
- (9) Is it possible to estimate the strain caused by the Ar bubbles from electron microscopy graphs and/or other methods to compare with the model calculations?
- (10) Can the explanation of the formation mechanism of the Ar bubbles and blisters help to control the sputtering process to energies that avoid the Ar implantation such as it might be in the case of CdS? Maybe also a change in the deposition time regime (pulse length, time between pulses) of the CdTe film growth could reduce the amount of Ar in the films.

Review form: Referee 2

Is the manuscript an original and important contribution to its field?

Good

Is the paper of sufficient general interest?

Good

Is the overall quality of the paper suitable?

Good

Can the paper be shortened without overall detriment to the main message?

Yes

Do you think some of the material would be more appropriate as an electronic appendix?

No

Do you have any ethical concerns with this paper?

No

Recommendation?

Accept with minor revision (please list in comments)

Comments to the Author(s)

The authors of present manuscript employ experimental and theoretical methods in order to assess the applicability of magnetron sputtering for more efficient thin film CdTe solar cell fabrication. The work is very interesting, in particular due the combination of experimental and theoretical investigations. It is therefore suitable for publication. However, there are some points which should be improved.

Page 5, 3 (a), second and fourth paragraph:

In paragraph 2 it is written that high-T CdCl₂ treatment leads to strong reduction of stacking faults and after the annealing the material has mainly zinc-blende structure. In paragraph 4, at first the thermal treatment without CdCl₂ is described, without mentioned whether mainly zinc-blende structure was formed or not, but small bubble formation is found. In the following sentence the authors mention once more high-T CdCl₂ treatment where micron sized bubbles are observed. I would propose to improve this part of the work and use clearer and statements and a better logics in the formulations.

Page 6, caption of Fig. 2:

"Fig. 3" should be certainly "Fig. (b)".

Page 6, caption of Fig. 3:

Typo: Should be "magnetron"

Page 9, Table 1 and related discussion:

Why is the 111 direction more open than the 110 direction? Compare with diamond structure which is very similar to zinc blende structure.

In the discussion of cluster formation in zinc-blende as well as in wurtzite I miss results concerning the total binding energy of a cluster and/or the binding energy of a single Ar or Xe atom to a cluster. Both binding energy and migration barriers are important characteristics to be considered in cluster formation.

In some cases figures are mentioned in the text too late, or the sequence of discussion in the text part does not agree with the sequence of figures. An example is on page 10 where the Arrhenius plot is mentioned but not Fig. 9 (Fig. 9 is later mentioned on page 11, after Fig. 10).

Page 10, bottom part:

The di-interstitial migration is discussed only very shortly, why?

Fig.9:

It should be mentioned which projection (view) is shown here, obviously it is onto a 110 plane. What is the symmetry of the (stable) interstitial(s)? I guess in figure (a) these are two types of tetrahedral interstitials at beginning and end of the jump.

Page 14, below Fig. 11: "Clusters are free...no evidence has been found of a maximum cluster size." This should be discussed in some more detail.

Fig. 14 shows a non-Arrhenius behaviour. Although not relevant for applications (because of high T), this interesting result should be discussed shortly.

Fig. 15 should be explained in more detail. Where is the boundary between wurtzite and zinc blende?

Comment: Fig. 16 shows an effect that one would expect without expensive simulations: The introduction of a high number of foreign atoms near the surface leads to an instability if the system is relaxed.

Page 16, first sentence of section 4: "is being sold at less than 0.03 per kWhr" What does this mean?

Decision letter (RSPA-2020-0056.R0)

01-Jun-2020

Dear Professor Smith,

On behalf of the Editor, I am pleased to inform you that your Manuscript RSPA-2020-0056 entitled "Inert Gas Bubble Formation in Magnetron Sputtered Thin Film CdTe Solar Cells" has been accepted for publication subject to minor revisions in Proceedings A. Please find the referees' comments below.

The reviewer(s) have recommended publication, but also suggest some minor revisions to your manuscript. Therefore, I invite you to respond to the reviewer(s)' comments and revise your manuscript. Please note that we have a strict upper limit of 28 pages for each paper. Please endeavour to incorporate any revisions while keeping the paper within journal limits. Please note that page charges are made on all papers longer than 20 pages. If you cannot pay these charges you must reduce your paper to 20 pages before submitting your revision. Your paper has been ESTIMATED to be 20 pages. We cannot proceed with typesetting your paper without your agreement to meet page charges in full should the paper exceed 20 pages when typeset. If you have any questions, please do get in touch.

It is a condition of publication that you submit the revised version of your manuscript within 7 days. If you do not think you will be able to meet this date please let me know in advance of the due date.

To revise your manuscript, log into <https://mc.manuscriptcentral.com/prsa> and enter your Author Centre, where you will find your manuscript title listed under "Manuscripts with Decisions." Under "Actions," click on "Create a Revision." Your manuscript number has been appended to denote a revision.

You will be unable to make your revisions on the originally submitted version of the manuscript. Instead, revise your manuscript and upload a new version through your Author Centre.

IMPORTANT: Your original files are available to you when you upload your revised manuscript. Please delete any redundant files before completing the submission process.

In addition to addressing all of the reviewers' and editor's comments, your revised manuscript **MUST** contain the following sections before the reference list (for any heading that does not apply to your work, please include a comment to this effect):

- Acknowledgements
- Funding statement

See <https://royalsociety.org/journals/authors/author-guidelines/> for further details.

When uploading your revised files, please make sure that you include the following as we cannot proceed without these:

- 1) A text file of the manuscript (doc, txt, rtf or tex), including the references, tables (including captions) and figure captions. Please remove any tracked changes from the text before submission. PDF files are not an accepted format for the "Main Document".
- 2) A separate electronic file of each figure (tif, eps or print-quality pdf preferred). The format should be produced directly from original creation package, or original software format.
- 3) Electronic Supplementary Material (ESM): all supplementary materials accompanying an accepted article will be treated as in their final form. Note that the Royal Society will not edit or typeset supplementary material and it will be hosted as provided. Please ensure that the supplementary material includes the paper details where possible (authors, article title, journal name). Supplementary files will be published alongside the paper on the journal website and posted on the online figshare repository (<https://figshare.com>). The heading and legend provided for each supplementary file during the submission process will be used to create the figshare page, so please ensure these are accurate and informative so that your files can be found in searches. Files on figshare will be made available approximately one week before the accompanying article so that the supplementary material can be attributed a unique DOI. Alternatively you may upload a zip folder containing all source files for your manuscript as described above with a PDF as your "Main Document". This should be the full paper as it appears when compiled from the individual files supplied in the zip folder.

Article Funder

Please ensure you fill in the Article Funder question on page 2 to ensure the correct data is collected for FundRef (<http://www.crossref.org/fundref/>).

Media summary

Please ensure you include a short non-technical summary (up to 100 words) of the key findings/importance of your paper. This will be used for to promote your work and marketing purposes (e.g. press releases). The summary should be prepared using the following guidelines:

*Write simple English: this is intended for the general public. Please explain any essential technical terms in a short and simple manner.

*Describe (a) the study (b) its key findings and (c) its implications.

*State why this work is newsworthy, be concise and do not overstate (true 'breakthroughs' are a rarity).

*Ensure that you include valid contact details for the lead author (institutional address, email address, telephone number).

Cover images

We welcome submissions of images for possible use on the cover of Proceedings A. Images should be square in dimension and please ensure that you obtain all relevant copyright permissions before submitting the image to us. If you would like to submit an image for consideration please send your image to proceedingsa@royalsociety.org

Once again, thank you for submitting your manuscript to Proceedings A and I look forward to receiving your revision. If you have any questions at all, please do not hesitate to get in touch.

Best wishes
Raminder Shergill
proceedingsa@royalsociety.org
Proceedings A

Reviewer(s)' Comments to Author:

Referee: 1

Comments to the Author(s)

The authors present an interesting work combining experimental investigations with simulations for a better understanding of the formation of Ar clusters and blisters during sputtering of CdTe films.

Minor changes and additional explanations are necessary prior to publication.

(1) In Figs. 2 - 5 electron microscopy graphs are shown. From my understanding there are TEM and also SEM images presented. Please specify in the Figure captions and add also the methods in Section 2 Methodology in more detail.

(2) The existence of the Ar clusters is based on the electron microscopy investigations. Especially TEM preparation with FIB or other techniques can result in contamination and artifacts with such very delicate gas structures. Did you check the effect of preparation on the formation of the Ar clusters? Are there analytical methods (other than microscopy) to prove the formation and existence of the Ar clusters?

(3) On p. 7 it is reported that 4% Ar is detected by EDS in the as-deposited films uniformly distributed over the whole CdTe layer of about 2-4 μm . However, if you compare with Fig. 2b the concentration of the Ar agglomeration in large internal voids and blisters seems to be higher than 4%. I guess Fig. 2b represents the cross section. Please give an explanation. A similar effect is also visible in Fig. 5 with blisters on the surface. The concentration seems to be larger than 4%. Can the Ar concentration be estimated directly from the images ?

(4) I also miss a quantitative analysis of the Ar clusters in size and size distribution from the experiment. The simulation model predicts interstitial type Ar clusters with larger ones pinned by replacement of lattice atoms by Ar. Both types are correlated with stress fields in the CdTe film. Have you measured stress in the film, and can this be predicted from the simulations maybe also with a variation of the Ar content?

(5) In the simulation model, it is reported that the most common Ar clusters after short time annealing are di-interstitials in the zinc-blende structure. Is there any explanation how they can form so favourably at elevated temperatures and how they are captured in the CdTe lattice structure?

(6) The formation of larger Ar clusters in the zinc-blende structure is correlated with the displacement of a lattice atom by an Ar atom when the cluster contains 6 or more Ar atoms. Then the Ar cluster is pinned. What would be the effect of a free surface in the simulation. Would you expect most of the Ar to diffuse out of the lattice in the case of a free surface in a purely zinc-blende system. Maybe this is why sputtered CdS exhibits no bubbles and it is only the pinning of Ar by the stacking faults that causes Ar clusters in CdTe. It would be interesting to undertake a simulation with a surface, with and without a stacking fault near the surface.

(7) In the wurtzite structure the energy barriers for diffusion are higher than in the zinc-blende structure and also 2 activation energies are detected (Fig. 14). For temperatures above 700 K the diffusion barrier is only 0.64 eV compared to 1.4 eV below 700 K. It is discussed that the Ar in a wurtzite state is more stable than in the zinc-blende structure since the lattice distortion by Ar is less. How can this be correlated with the 2 activation energies observed in wurtzite and the much lower activation energy at elevated temperatures?

(8) How big are Ar clusters in the wurtzite structure? It seems that the Ar atoms are less mobile at least at low temperature. Is the low mobility related only to interstitial Ar clusters or is there also displacement of lattice atoms by Ar atoms observed?

(9) Is it possible to estimate the strain caused by the Ar bubbles from electron microscopy graphs and/or other methods to compare with the model calculations?

(10) Can the explanation of the formation mechanism of the Ar bubbles and blisters help to control the sputtering process to energies that avoid the Ar implantation such as it might be in the case of CdS? Maybe also a change in the deposition time regime (pulse length, time between pulses) of the CdTe film growth could reduce the amount of Ar in the films.

Referee: 2

Comments to the Author(s)

The authors of present manuscript employ experimental and theoretical methods in order to assess the applicability of magnetron sputtering for more efficient thin film CdTe solar cell fabrication. The work is very interesting, in particular due the combination of experimental and theoretical investigations. It is therefore suitable for publication. However, there are some points which should be improved.

Page 5, 3 (a), second and fourth paragraph:

In paragraph 2 it is written that high-T CdCl₂ treatment leads to strong reduction of stacking faults and after the annealing the material has mainly zinc-blende structure. In paragraph 4, at first the thermal treatment without CdCl₂ is described, without mentioned whether mainly zinc-blende structure was formed or not, but small bubble formation is found. In the following sentence the authors mention once more high-T CdCl₂ treatment where micron sized bubbles are observed. I would propose to improve this part of the work and use clearer and statements and a better logics in the formulations.

Page 6, caption of Fig. 2:

“Fig. 3” should be certainly “Fig. (b)”.

Page 6, caption of Fig. 3:

Typo: Should be “magnetron”

Page 9, Table 1 and related discussion:

Why is the 111 direction more open than the 110 direction? Compare with diamond structure which is very similar to zinc blende structure.

In the discussion of cluster formation in zinc-blende as well as in wurtzite I miss results concerning the total binding energy of a cluster and/or the binding energy of a single Ar or Xe atom to a cluster. Both binding energy and migration barriers are important characteristics to be considered in cluster formation.

In some cases figures are mentioned in the text too late, or the sequence of discussion in the text part does not agree with the sequence of figures. An example is on page 10 where the Arrhenius plot is mentioned but not Fig. 9 (Fig. 9 is later mentioned on page 11, after Fig. 10).

Page 10, bottom part:

The di-interstitial migration is discussed only very shortly, why?

Fig.9:

It should be mentioned which projection (view) is shown here, obviously it is onto a 110 plane. What is the symmetry of the (stable) interstitial(s)? I guess in figure (a) these are two types of tetrahedral interstitials at beginning and end of the jump.

Page 14, below Fig. 11: "Clusters are free....no evidence has been found of a maximum cluster size." This should be discussed in some more detail.

Fig. 14 shows a non-Arrhenius behaviour. Although not relevant for applications (because of high T), this interesting result should be discussed shortly.

Fig. 15 should be explained in more detail. Where is the boundary between wurtzite and zinc blende?

Comment: Fig. 16 shows an effect that one would expect without expensive simulations: The introduction of a high number of foreign atoms near the surface leads to an instability if the system is relaxed.

Page 16, first sentence of section 4: "is being sold at less than 0.03 per kWhr" What does this mean?

Author's Response to Decision Letter for (RSPA-2020-0056.R0)

See Appendix A.

Decision letter (RSPA-2020-0056.R1)

24-Jun-2020

Dear Professor Smith

I am pleased to inform you that your manuscript entitled "Inert Gas Bubble Formation in Magnetron Sputtered Thin Film CdTe Solar Cells" has been accepted in its final form for publication in Proceedings A.

Our Production Office will be in contact with you in due course. You can expect to receive a proof of your article soon. Please contact the office to let us know if you are likely to be away from e-

mail in the near future. If you do not notify us and comments are not received within 5 days of sending the proof, we may publish the paper as it stands.

Open access

You are invited to opt for open access, our author pays publishing model. Payment of open access fees will enable your article to be made freely available via the Royal Society website as soon as it is ready for publication. For more information about open access please visit http://royalsocietypublishing.org/site/authors/open_access.xhtml. The open access fee for this journal is £1700/\$2380/€2040 per article. VAT will be charged where applicable.

Note that if you have opted for open access then payment will be required before the article is published – payment instructions will follow shortly.

If you wish to opt for open access then please inform the editorial office (proceedingsa@royalsociety.org) as soon as possible.

Your article has been estimated as being 20 pages long. Our Production Office will inform you of the exact length at the proof stage.

Proceedings A levies charges for articles which exceed 20 printed pages. (based upon approximately 540 words or 2 figures per page). Articles exceeding this limit will incur page charges of £150 per page or part page, plus VAT (where applicable).

Under the terms of our licence to publish you may post the author generated postprint (ie. your accepted version not the final typeset version) of your manuscript at any time and this can be made freely available. Postprints can be deposited on a personal or institutional website, or a recognised server/repository. Please note however, that the reporting of postprints is subject to a media embargo, and that the status the manuscript should be made clear. Upon publication of the definitive version on the publisher's site, full details and a link should be added.

You can cite the article in advance of publication using its DOI. The DOI will take the form: 10.1098/rspa.XXXX.YYYY, where XXXX and YYYY are the last 8 digits of your manuscript number (eg. if your manuscript number is RSPA-2017-1234 the DOI would be 10.1098/rspa.2017.1234).

For tips on promoting your accepted paper see our blog post: <https://blogs.royalsociety.org/publishing/promoting-your-latest-paper-and-tracking-your-results/>

On behalf of the Editor of Proceedings A, we look forward to your continued contributions to the Journal.

Sincerely,
Raminder Shergill
proceedingsa@royalsociety.org

Appendix A

Reviewer(s)' Comments to Author:

Referee: 1

Comments to the Author(s)

The authors present an interesting work combining experimental investigations with simulations for a better understanding of the formation of Ar clusters and blisters during sputtering of CdTe films.

Minor changes and additional explanations are necessary prior to publication.

(1) In Figs. 2 – 5 electron microscopy graphs are shown. From my understanding there are TEM and also SEM images presented. Please specify in the Figure captions and add also the methods in Section 2 Methodology in more detail.

Response:

Figure 2 caption: Introduction has been modified to read “Low resolution cross-sectional TEM images of a CdTe device a)...”

Figure 3 caption: Introduction modified to read “A High Resolution TEM image of a cross-section of an as-deposited CdTe device deposited....”

Figure 4 caption: modified to read “TEM images of the structure....”

Figure 5 caption: modified to read “SEM images of blisters...”

We have also added text in the methodology to explain how the SEM images were obtained :

“Samples for Transmission Electron Microscopy (TEM) were prepared by an in-situ lift out method using a FEI Nova 600 Nanolab Focused Ion Beam (FIB), which was also equipped with an in-lens SEM detector used for detailed surface imaging”.

(2) The existence of the Ar clusters is based on the electron microscopy investigations. Especially TEM preparation with FIB or other techniques can result in contamination and artifacts with such very delicate gas structures. Did you check the effect of preparation on the formation of the Ar clusters? Are there analytical methods (other than microscopy) to prove the formation and existence of the Ar clusters?

Response: This is a common question. However, we are very confident that the ion beam milling used to prepare the TEM samples has not caused artifacts that affect our interpretation. The Ga ion beam is used at a shallow angle to minimize any damage to the outer surface atomic layers and these layers are then removed by a low-energy ion etch. We have processed literally hundreds of device structures in this way without artifact issues arising. The Ar clusters observed after annealing without the presence of chlorine are typically 5nm in diameter and are probably too small for any technique other than HRTEM.

(3) On p. 7 it is reported that 4% Ar is detected by EDS in the as-deposited films uniformly distributed over the whole CdTe layer of about 2-4 μm . However, if you compare with Fig. 2b the concentration of the Ar agglomeration in large internal voids and blisters seems to be higher than 4%. I guess Fig. 2b represents the cross section. Please give an explanation. A similar effect is also visible in Fig. 5 with blisters on the surface. The concentration seems to be larger than 4%. Can the Ar concentration be estimated directly from the images ?

Response: Estimation of the Argon percentage from the images is hard if not impossible since, for example, Fig 2b is a cross-sectional image chosen specifically to show the large elongated voids caused by Ar agglomeration rather than a representation of all such cross sections therefore it may not represent the 'average' cross section. With regards to Fig. 5 what we are seeing here is a result of the Ar clustering rather than a direct representation of the cluster sizes and therefore percentage of Ar in the film. The only realistic measurement which can be made to estimate the concentration of Ar in the film which would be useful for simulation purposes is to detect the Ar in the as-deposited film through EDS.

(4) I also miss a quantitative analysis of the Ar clusters in size and size distribution from the experiment. The simulation model predicts interstitial type Ar clusters with larger ones pinned by replacement of lattice atoms by Ar. Both types are correlated with stress fields in the CdTe film. Have you measured stress in the film, and can this be predicted from the simulations maybe also with a variation of the Ar content?

Response: This is a very good question! Our initial thoughts were that the blister formation was linked to stress in the films. We measured the stress using XRD in the as-deposited films and modified the process gas pressures to minimise the deposition energy. The results were published in:

‘Internal strain analysis of CdTe thin films deposited by pulsed DC magnetron sputtering’ Piotr M Kaminski, Ali Abbas, C Chen, Sibel Yilmaz, Francesco Bittau, Jake W Bowers, JM Walls, IEEE 42nd Photovoltaic Specialist Conference (PVSC) (2015) 1-6 . This paper is reference 14.

To emphasise the point and draw the reader's attention to this previous work the sentence at line 54 has been extended and now reads 'This temperature and the use of relatively high working gas pressure with gas flows of 50sccm has been found to minimize stress in the films'

In fact, the stress is not particularly high in the as-deposited material. However, once the CdCl₂ treatment is used, the stacking faults are removed, and the argon is free to diffuse and form bubbles which causes blistering. The blistering releases the stress caused by the bubble formation. The process of bubble/blister formation is very fast

(5) In the simulation model, it is reported that the most common Ar clusters after short time annealing are di-interstitials in the zinc-blende structure. Is there any explanation how they can form so favourably at elevated temperatures and how they are captured in the CdTe lattice structure?

Response: Explanation added to text: "The annealing simulations show that a large number of di-interstitial clusters form during the clustering process. This is due to a low barrier of 0.2 eV for a di-interstitial to form providing two Ar/Xe atoms are in adjacent interstitial sites. The elevated reverse barrier of 0.9 eV for the dissipation of the di-interstitial also contributes to the large number of these defects [35]."

For the reviewer's convenience, the cited paper is: [35] Hatton P, Goddard P, Smith R, Abbas A, Potiamalis C, Greenhalgh R, Walls JM. 2019 Inert gas cluster formation in sputter-deposited thin film CdTe solar cells. Thin Solid Films 692 137614.

(6) The formation of larger Ar clusters in the zinc-blende structure is correlated with the displacement of

a lattice atom by an Ar atom when the cluster contains 6 or more Ar atoms. Then the Ar cluster is pinned. What would be the effect of a free surface in the simulation. Would you expect most of the Ar to diffuse out of the lattice in the case of a free surface in a purely zinc-blende system. Maybe this is why sputtered CdS exhibits no bubbles and it is only the pinning of Ar by the stacking faults that causes Ar clusters in CdTe. It would be interesting to undertake a simulation with a surface, with and without a stacking fault near the surface.

Response: The simulation discussed here has been undertaken by the authors previously and it was found that the effect of the free surface was minimal allowing only Ar/Xe in the top two layers to diffuse out of the system. Below that the Ar/Xe will not feel the effects of the free surface and therefore the diffusion will be random until clusters begin to form as described in this piece of work. The mechanisms of cluster growth, or lack thereof, in CdS is currently being investigated by the authors.

Explanation of this added to text: "Free surface calculations have also been conducted in which one of surfaces' periodic boundary conditions were removed and atoms were able to flow out of the CdTe. Upon heating, in cases described previously, we found that only Ar/Xe atoms in the top 2 layers were able to escape, below that the effect of the free surface was not felt and diffusion was mainly into clusters as discussed above."

(7) In the wurtzite structure the energy barriers for diffusion are higher than in the zinc-blende structure and also 2 activation energies are detected (Fig. 14). For temperatures above 700 K the diffusion barrier is only 0.64 eV compared to 1.4 eV below 700 K. It is discussed that the Ar in a wurtzite state is more stable than in the zinc-blende structure since the lattice distortion by Ar is less. How can this be correlated with the 2 activation energies observed in wurtzite and the much lower activation energy at elevated temperatures?

Note: The authors noticed an error in the text, the change in gradient noted to occur at 700 K actually occurs at 800 K and this has been changed in the text.

Response: The reviewer raises an interesting question about the nature of the energy landscape in the high temperature scenario of Fig 14. This finding of non-Arrhenius behaviour is a rare finding and to the authors knowledge, no tool is available to be able to investigate the reviewer's question. We cannot know the effect of the high temperature on the saddle point nor the stable site energy. All we can know, using Fig. 14, is that the diffusion barrier will decrease to ~0.64 eV at these temperatures but below these temperatures the barrier will be ~1.4 eV. We can investigate the low temperature cases since we have tools such as NEB which operate at effectively 0 K.

(8) How big are Ar clusters in the wurtzite structure? It seems that the Ar atoms are less mobile at least at low temperature. Is the low mobility related only to interstitial Ar clusters or is there also displacement of lattice atoms by Ar atoms observed?

Response: Ar in a wurtzite structure is not able to displace lattice atoms, at least not alone. Perhaps a cluster of Ar atoms could, in wurtzite, displace lattice atoms but due to the high barriers for diffusion in wurtzite, clusters will not form at the experimental temperature considered in this work.

(9) Is it possible to estimate the strain caused by the Ar bubbles from electron microscopy graphs and/or other methods to compare with the model calculations?

Response: Please see our response to point 4 above. The strain in sputtered CdTe films has been considered in Reference 14 'Internal strain analysis of CdTe thin films deposited by pulsed DC magnetron sputtering' Piotr M Kaminski, Ali Abbas, C Chen, Sibel Yilmaz, Francesco Bittau, Jake W Bowers, JM Walls, IEEE 42nd Photovoltaic Specialist Conference (PVSC) (2015) 1-6 .

In practice, bubble formation is caused by the CdCl₂ process and is linked to the removal of stacking faults. Blistering occurs almost simultaneously causing the stress/strain to be relieved.

(10) Can the explanation of the formation mechanism of the Ar bubbles and blisters help to control the sputtering process to energies that avoid the Ar implantation such as it might be in the case of CdS? Maybe also a change in the deposition time regime (pulse length, time between pulses) of the CdTe film growth could reduce the amount of Ar in the films.

Response: We thought along the same lines and refer to the ideas for reducing inert gas implantation in Section 4. We have had some success with switching to Xe as the working gas, but this seems to be dependent on the quality of the magnetron target. Unfortunately, the greater the impurity content, the lower the Voltage! Lowering the target Voltage lowers the energy of deposition and results in blister free devices. Unfortunately, the impurities in the film are then detrimental to the device.

Referee: 2

Comments to the Author(s)

The authors of present manuscript employ experimental and theoretical methods in order to assess the applicability of magnetron sputtering for more efficient thin film CdTe solar cell fabrication. The work is very interesting, in particular due the combination of experimental and theoretical investigations. It is therefore suitable for publication. However, there are some points which should be improved.

(1) Page 5, 3 (a), second and fourth paragraph:

In paragraph 2 it is written that high-T CdCl₂ treatment leads to strong reduction of stacking faults and after the annealing the material has mainly zinc-blende structure. In paragraph 4, at first the thermal treatment without CdCl₂ is described, without mentioned whether mainly zinc-blende structure was formed or not, but small bubble formation is found. In the following sentence the authors mention once more high-T CdCl₂ treatment where micron sized bubbles are observed. I would propose to improve this part of the work and use clearer and statements and a better logics in the formulations.

Response: The authors agree that this section may have been confusing. Therefore, we have reordered the paragraphs to a more logical ordering and clarified the points raised by the reviewer within the text.

(2) Page 6, caption of Fig. 2:

“Fig. 3” should be certainly “Fig. (b)”.

Response: Corrected typo

(3) Page 6, caption of Fig. 3:
Typo: Should be “magnetron”

Response: Corrected typo

(4) Page 9, Table 1 and related discussion:

Why is the 111 direction more open than the 110 direction? Compare with diamond structure which is very similar to zinc blende structure.

Response: Although (110) has long channels which run parallel to the implantation trajectory the actual structure on the top layer of (110) is more densely packed with atoms meaning there is less space for an incoming atom to implant into those gaps. In a (111) structure there are larger gaps in atoms on the top layer meaning there is more opportunity for implantation since the incoming atom has a better chance at getting beneath the top layer and therefore implanting into the film.

This has been clarified in the text and the line now reads: “Also, penetration depends heavily on surface orientation, with (1 1 1) allowing for the most penetration of both Ar and Xe due to the open structure of its top layer compared with other orientations.”

(5) In the discussion of cluster formation in zinc-blende as well as in wurtzite I miss results concerning the total binding energy of a cluster and/or the binding energy of a single Ar or Xe atom to a cluster. Both binding energy and migration barriers are important characteristics to be considered in cluster formation.

Response: The authors agree that some findings related to binding energy would be useful in assessing cluster dissipation, or lack thereof, in zinc blende CdTe. Therefore, some discussion of binding energy has been added in a paragraph just before the Wurtzite subsection which reads:

“We have demonstrated above that the dissipation of Ar clusters will be rare due to large barriers for atoms to leave a cluster. The binding energy of Ar/Xe atoms to their respective clusters will help indicate in general how likely clusters of different sizes are to disassemble. In small clusters of size 2-5 atoms the binding energy of a single atom to the cluster is between ~ -1.1 eV and ~ -1.3 eV for Ar and Xe. For larger clusters of 10 atoms the binding energies are ~ -1.8 eV for both Ar and Xe. For larger clusters of around 30 atoms in size, the binding energy is ~ -2.9 eV for both Ar and Xe. These binding energies are relatively large when compared with diffusion barriers of these atoms in a bulk system and are therefore not likely to be overcome under experimental temperatures which will restrict the likelihood of dissipation of clusters even as small as 5 atoms.”

In terms of wurtzite cluster binding energy, we have concluded in this paper that clusters will not form in wurtzite due to the large energy barrier for diffusion so this was not considered.

(6) In some cases figures are mentioned in the text too late, or the sequence of discussion in the text part does not agree with the sequence of figures. An example is on page 10 where the Arrhenius plot is mentioned but not Fig. 9 (Fig. 9 is later mentioned on page 11, after Fig. 10).

Response: Some figures have been reordered to match sequence of discussion

(7) Page 10, bottom part:

The di-interstitial migration is discussed only very shortly, why?

Response: although di-interstitials form readily during annealing, there is no true migration mechanism for the di-interstitial. It is instead dictated by migration of solo Ar/Xe atoms through the mentioned 'break away, catch-up' mechanism. However, the barrier of these transitions are >0.9 eV and have low reverse barriers so were determined of to be low significance.

In response to another reviewers' comments we have added further explanation of how di-interstitials form which might help with understanding the di-interstitials further:

"The annealing simulations show that a large number of di-interstitial clusters form during the clustering process. This is due to a low barrier of 0.2 eV for a di-interstitial to form providing two Ar/Xe atoms are in adjacent interstitial sites. The elevated reverse barrier of 0.9 eV for the dissipation of the di-interstitial also contributes to the large number of these defects [35]."

For the reviewer's convenience, the cited paper is: [35] Hatton P, Goddard P, Smith R, Abbas A, Potiamalis C, Greenhalgh R, Walls JM. 2019 Inert gas cluster formation in sputter-deposited thin film CdTe solar cells. *Thin Solid Films* 692 137614.

(8) Fig.9:

It should be mentioned which projection (view) is shown here, obviously it is onto a 110 plane.

What is the symmetry of the (stable) interstitial(s)? I guess in figure (a) these are two types of tetrahedral interstitials at beginning and end of the jump.

Response: Presuming this was in reference to Fig 8 rather than Fig 9.

The figure caption has been updated to reflect the (110) plane view and the fact that the transition is between the two tetrahedral sites:

"Various diffusion mechanisms for a single Ar interstitial. (a) Direct transformation from a Cd-coordinated interstitial site to a Te-coordinated site with an energy barrier of 0.6 eV shown onto the (110) plane..."

Note: As a result of this comment the authors noticed the arrow in Fig 8a) was pointing in the wrong direction, this has now been fixed.

(9) Page 14, below Fig. 11: "Clusters are free....no evidence has been found of a maximum cluster size." This should be discussed in some more detail.

Response: The sentence in question has now been expanded to read: "Clusters are free to grow and further investigation has not found a cluster size which inhibits the mechanisms for cluster growth. Moreover, even clusters of 53 atoms encourage Ar/Xe incorporation through reduced barriers for faster diffusion into the cluster and energy decreases once the atom has joined. Therefore, we have found no evidence for a maximum cluster size in bulk zinc-blende oriented CdTe."

(10) Fig. 14 shows a non-Arrhenius behaviour. Although not relevant for applications (because of high T), this interesting result should be discussed shortly.

Response: The authors agree with the reviewer's comment and have added a short paragraph to discuss the significance of this finding which reads:

“This non-Arrhenius behaviour is a rare finding, especially in solid state modelling, where large structural changes are not occurring. As a result of this behaviour, any form of traditional temperature assisted dynamics where the user crosses 800 K would result in findings that would not be representative of results at the original temperature. In many systems, increasing the temperature to simulate transitions faster is a valid method as it is assumed, sometimes correctly, that the Arrhenius plot is linear, as in Fig. 9, however, as we have shown here it is important to check”.

(11) Fig. 15 should be explained in more detail. Where is the boundary between wurtzite and zinc blende?

Response: Caption updated to include more detail: “NEB calculation of Ar diffusion from a zinc-blende structure (Ar positions 1 and 2), formed by the top two layers of CdTe, into a wurtzite structure (Ar positions 3 and 4) which is induced by the stacking fault in layer 3. A large energy decrease is associated to the transition of Ar from zinc-blende into wurtzite due to the open structure of wurtzite. This results in Ar requiring a ~ 1.2 eV barrier for either diffusion which means the Ar is effectively trapped at the stacking fault.”

(12) Comment: Fig. 16 shows an effect that one would expect without expensive simulations: The introduction of a high number of foreign atoms near the surface leads to an instability if the system is relaxed.

Response: The authors agree with the reviewer's comment but feel the images are valuable for illustrative purposes and as a comparison to experimental images.

(13) Page 16, first sentence of section 4: “is being sold at less than 0.03 per kWhr” What does this mean?

Response: This refers to the cost of electricity from thin film CdTe solar utilities which is below \$0.03/kWhr. The dollar sign has been added in the text.